# Unveiling the Exquisite Microstructural Details in Zebrafish Brain Non-Invasively Using Magnetic Resonance Imaging at 28.2 T

**DOI:** 10.3390/molecules29194637

**Published:** 2024-09-29

**Authors:** Rico Singer, Ina Oganezova, Wanbin Hu, Yi Ding, Antonios Papaioannou, Huub J. M. de Groot, Herman P. Spaink, A Alia

**Affiliations:** 1Leiden Institute of Chemistry, Leiden University, Einsteinweg 55, 2301 RA Leiden, The Netherlands; r.singer@lic.leidenuniv.nl (R.S.); i.oganezova@umail.leidenuniv.nl (I.O.); groot_h@lic.leidenuniv.nl (H.J.M.d.G.); 2Institute of Biology, Leiden University, Einsteinweg 55, 2301 RA Leiden, The Netherlands; w.hu@biology.leidenuniv.nl (W.H.); y.ding@biology.leidenuniv.nl (Y.D.); h.p.spaink@biology.leidenuniv.nl (H.P.S.); 3Bruker BioSpin GmbH, NMR Microscopy, 76287 Rheinstetten, Germany; antonios.papaioannou@bruker.com; 4Institut für Medizinische Physik und Biophysik, Universität Leipzig, Härtelstr. 16-18, D-04107 Leipzig, Germany

**Keywords:** magnetic resonance imaging, diffusion MRI, white matter tractography, zebrafish

## Abstract

Zebrafish (*Danio rerio*) is an important animal model for a wide range of neurodegenerative diseases. However, obtaining the cellular resolution that is essential for studying the zebrafish brain remains challenging as it requires high spatial resolution and signal-to-noise ratios (SNR). In the current study, we present the first MRI results of the zebrafish brain at the state-of-the-art magnetic field strength of 28.2 T. The performance of MRI at 28.2 T was compared to 17.6 T. A 20% improvement in SNR was observed at 28.2 T as compared to 17.6 T. Excellent contrast, resolution, and SNR allowed the identification of several brain structures. The normative *T*_1_ and *T*_2_ relaxation values were established over different zebrafish brain structures at 28.2 T. To zoom into the white matter structures, we applied diffusion tensor imaging (DTI) and obtained axial, radial, and mean diffusivity, as well as fractional anisotropy, at a very high spatial resolution. Visualisation of white matter structures was achieved by short-track track-density imaging by applying the constrained spherical deconvolution method (stTDI CSD). For the first time, an algorithm for stTDI with multi-shell multi-tissue (msmt) CSD was tested on zebrafish brain data. A significant reduction in false-positive tracks from grey matter signals was observed compared to stTDI with single-shell single-tissue (ssst) CSD. This allowed the non-invasive identification of white matter structures at high resolution and contrast. Our results show that ultra-high field DTI and tractography provide reproducible and quantitative maps of fibre organisation from tiny zebrafish brains, which can be implemented in the future for a mechanistic understanding of disease-related microstructural changes in zebrafish models of various brain diseases.

## 1. Introduction

Zebrafish (*Danio rerio*) have become an excellent animal model in the studies of diseases, biological pathways, genetics, and toxicology [1,2,3,4]. In the field of neurodegenerative conditions, zebrafish offer various models, including those for Alzheimer’s, Parkinson’s, and Huntington’s disease [5,6,7]. Therefore, studying the zebrafish brain non-invasively might provide valuable information on the pathology and treatment of neurodegenerative disorders. Magnetic resonance imaging (MRI) is a well-established, non-invasive technique for neuroimaging in both human and animal models. In our previous reports, a successful examination of zebrafish was performed at high field (9.4 T) [8,9,10] and ultra-high field (17.6 T) MRI [11,12,13]. High-quality images gave access to anatomical details, allowing visualisation of white matter (WM) lesions in zebrafish models of familial cystic leukoencephalopathy [9] and Lowe syndrome [10], as well as in vivo analysis of malignant melanoma tumours [13]. Additionally, in vivo high-resolution localised magnetic resonance spectroscopy (MRS) was successfully applied to obtain the neurochemical metabolite profile of adult zebrafish. However, obtaining the essential resolution for studying small structures with a high signal-to-noise ratio (SNR) remains challenging. 

Insight into the microstructural organisation of the zebrafish brain could be obtained by diffusion-based MRI (dMRI), a powerful, non-invasive technique with high sensitivity for water movement [14]. Cellular structures hinder the microscopic random motion of water, making dMRI unique in studying the microstructural organisation of tissue. Diffusion tensor imaging (DTI) is an extended dMRI method providing increased structural information by exploiting anisotropic diffusion effects. Diffusion tensors are calculated from directional differences in the MR signals and used to determine the axial diffusivity (*D*_∥_) from the principal eigenvalue, the radial diffusivity (*D*_⊥_) from the average of two non-principal eigenvalues, the mean diffusivity (*MD*), and the fractional anisotropy (*FA*), the extent of directional preference. In the brain, anisotropic diffusion effects are most prominent in WM due to the ordered structures of its myelinated axon tracts [14]. Changes in the diffusion anisotropy of WM structures have been reported for many neurodegenerative diseases, including Alzheimer’s disease, Parkinson’s disease, Amyotrophic Lateral Sclerosis (ALS), and Huntington’s Disease [15,16,17,18,19,20,21,22,23,24,25]. The directionality of WM structures is visualised by directionally encoded colour (DEC) *FA* maps, combining *FA* with the directionality of the principal eigenvector. However, *FA* colour maps do not visualise the connectivity of WM tracks, nor does the technique account for crossing or closely passing fibres below the applied MRI resolution. In this regard, DTI offers a solution for visualising WM tracks. DTI tractography is a distinct processing technique of DTI tensors and the only known non-invasive imaging technique for visualising WM connectivity in the brain. DTI has been successfully used to probe the changes in brain connectivity during neurogenerative diseases in human subjects [26,27,28,29,30,31,32]. Combined with the high spatial resolution required for the neurological analysis of zebrafish, dMRI is very challenging for the zebrafish brain. Consequently, knowledge of diffusivity and connectivity in the zebrafish brain is limited. Freidlin et al. [33] obtained good contrast of the spinal cord in adult zebrafish by DTI, while Ullmann et al. [34] presented a DTI study in the zebrafish brain and obtained tractography maps using short-track track density imaging with single-shell single-tissue constrained spherical deconvolution (stTDI ssst-CSD). However, the analysis was performed on isolated brain tissue rather than intact zebrafish. Additionally, Ullmann et al. performed DTI with a single non-zero *b*-value, eliminating the possibility of individually estimating WM, grey matter (GM), and cerebrospinal fluid (CSF) signals, therefore possibly resulting in errors and WM overestimation during tractography [35]. Recently, multi-shell multi-tissue (msmt) CSD algorithms were developed, deconvoluting WM, GM, and CSF responses [35]. By filtering GM- and CSF-like signals strongly present in ssst-CSD, false positive tracks are reduced [36,37]. We have recently utilised msmt CSD methods at 17.6 T to identify white matter structures in the zebrafish brain of a Toll-like receptor 2 deficient zebrafish model, aiming to compare white matter integrity [38]. However, a comparison of ssst CSD and msmt-CSD has not been validated for the zebrafish brain. In the current work, ssst CSD and msmt CSD methods are compared, marking a critical step in adapting and validating these advanced imaging techniques for use in zebrafish neuroimaging.

Exploiting higher magnetic fields for imaging can potentially improve the neurological analysis of zebrafish brains as SNR increases with the applied magnetic field strength (*B*_0_). Consequently, increased spatial resolution can be obtained without the need for significant elongation of total acquisition time. In this study, the first MRI results at 28.2 T are presented that were obtained from the zebrafish brain. The performance of MRI at 28.2 T was compared to 17.6 T, and a wide range of MR sequences were optimised, including anatomical imaging by rapid acquisition with relaxation enhancement (RARE). Moreover, diffusion weighted imaging (DWI) was applied to quantify apparent diffusion coefficient (*ADC*) values in several zebrafish brain regions. Furthermore, white matter tractography was conducted through DTI using stTDI CSD. Our findings not only include the initial results of stTDI through ssst-CSD on intact zebrafish but also show the first outcomes of stTDI through msmt-CSD on the zebrafish brain.

## 2. Results and Discussion

This study presents the first MRI findings of the zebrafish brain using a state-of-the-art magnetic field strength of 28.2 T. The zebrafish has become a popular species for studying neurological disorders due to the growing understanding of its nervous system and the availability of numerous transgenic zebrafish models [5,6,9,10,13]. Non-invasive MRI methods have significant potential for investigating brain pathology in these models. However, obtaining the essential resolution for studying microstructures and diffusion processes with a high SNR remains challenging for zebrafish brains. In this work, we have optimised and successfully applied MRI methods at 28.2 T magnetic field strength to resolve microstructural details and white matter tracts in the young adult zebrafish brain.

### 2.1. Relaxation Times 

The application of MRI at an ultra-high field of 28.2 T requires adjustment of image acquisition parameters, which are based on knowledge of the MR relaxation properties of the tissues being imaged. To establish regional *T*_1_ and *T*_2_ values of the zebrafish brain at 28.2 T, relaxation times were estimated in nine manually selected ROIs (Figure 1A): the vagal lobe (Xlo), granular layer of the cerebellar corpus (CeG), telencephalon (Te), olfactory bulb (OB), molecular layer of the cerebellar corpus (Cem), cerebellar crest (CC), optic tectum (OT), caudal zone of the periventricular hypothalamus (Hc), and the ventral zone of the periventricular hypothalamus (Hv). Figure 1B shows relaxation times *T*_1_ and *T*_2_ for selected ROIs at 28.2 T and compares them with values obtained at 17.6 T. A clear increase in *T*_1_ and a decrease in *T*_2_ were observed in all selected ROIs at 28.2 T compared to 17.6 T. On average, *T*_1_ increased by a factor of 1.097 ± 0.015, while *T*_2_ decreased by a factor of 0.905 ± 0.011 at 28.2 T compared to 17.6 T. The observed shift in relaxation times at increasing *B*_0_ is consistent with previous reports [39,40,41]. Previously, *T*_1_ was found to be proportional to *B*_0_^1/3^ [42]. However, some studies suggest a linear increase in *T*_1_ with *B*_0_ [43]. In our findings, *T*_1_ increased in all brain regions but at a rate lower than the theoretical *B*_0_^1/3^ expectation [42,43]. This is consistent with *T*_1_ reported in mouse brain tissue at 9.4 T and 17.6 T, where an average *T*_1_ increase factor of 1.08 was reported, well below the theoretical increase factor of 1.23 expected under a *B*_0_^1/3^ dependence [44]. For *T*_2_, the observed decrease at higher magnetic fields is consistent with previous reports as well. In mouse brain regions, increasing the magnetic field from 9.4 T to 17.6 T resulted in an average *T*_2_ decrease by a factor of 0.72 [45], changing from 9.4 T to 11.7 T resulted in an average *T*_2_ decrease by a factor of 0.83 [40]. Besides *B*_0_, the relaxation times of tissues depend on factors like age, gender, acquisition parameters, and pre-treatment (measuring in vivo or fixed) [46]. To minimise the effects of additional factors, measurements at 17.6 T and 28.2 T were performed for the same zebrafish with identical acquisition parameters, including *TE*, *TR*, RARE factor, number of averages, and acquisition bandwidth. Furthermore, both MR systems contained similar RF coils, gradient power supplies, and software. To summarise, we have established the regional *T*_1_ and *T*_2_ values of the healthy zebrafish brain at 28.2 T. Our findings indicate that *T*_2_ decreases while *T*_1_ continues to rise with increasing magnetic field strength up to the ultra-high 28.2 T level. In future studies, accurate knowledge of *T*_1_ and *T*_2_ values of various brain regions at 28.2 T can serve as a reference point for detecting regional relaxation changes related to diseases in zebrafish brains. 

### 2.2. Anatomical Imaging

Figure 2 shows representative slices of images of the zebrafish brain acquired using the RARE sequence at 17.6 T and 28.2 T. Images were acquired using the same zebrafish samples and identical acquisition parameters at both magnetic field strengths, obtaining a resolution of 23 μm × 23 μm and a slice thickness of 200 μm. A clear improvement in SNR is observed at 28.2 T as compared to 17.6 T (Figure 2A). On average, an SNR improvement factor of 1.2 was found at 28.2 T as compared to 17.6 T in the zebrafish brain. Several brain structures could be clearly identified at 28.2 T (Figure 2B). Figure 2B shows anatomical images obtained at 28.2 T in the coronal and sagittal direction at a field of view (FOV) of 6 × 6 mm, an image size of 256 × 256 voxels, and a slice thickness of 100 μm, resulting in a spatial resolution of 23 × 23 × 100 μm. Excellent contrast, resolution, and SNR allowed for the identification of several brain structures that were verified by comparing them with detailed atlases of the zebrafish brain [47,48].

### 2.3. Diffusion Weighted Imaging (DWI)

To obtain further contrast for the identification of WM structures such as commissures, fibre tracts, nerves, as well as the CSF system, we applied diffusion-based MRI techniques. dMRI provides contrast based on the Brownian motion of water molecules [49]. MRI is made sensitive to diffusion by strong gradient pulses before and after a 180° refocusing pulse, of which the magnitude (G), duration (δ), and time interval (∆) are summarised in the *b*-value. In Figure 3A, the effect of the applied *b*-value in DWI experiments is shown for the zebrafish brain. At low *b*-values of 75 s/mm^2^, high contrast and SNR are obtained for all regions in the zebrafish head. At moderate *b*-values of 500 and 1000 s/mm^2^, the signal intensity of muscle and surrounding tissue is reduced, while a high contrast and SNR in the brain is maintained. In fact, DWI images obtained at 1000 s/mm^2^ show better contrast between various brain regions than those obtained at 75 and 500 s/mm^2^. This is clearer in the forebrain, where the olfactory bulb gained significant contrast compared to anatomical imaging (Figure 2B and Figure 3A). This allowed better contrast for the identification of various structures in the brain compared to anatomical images obtained by the RARE sequence (Appendix A). At a high *b*-value of 3500 s/mm^2^, the MR signal of all tissue drops, although the brain is still visible. Some signals from the brain were even visible at very high *b*-values of 5000 and 7500 s/mm^2^. Figure 3B shows the *ADC* map. To eliminate the effect of non-Gaussian diffusion effects, the MR signals obtained at high *b*-values (>2000 s/mm^2^) were excluded during the estimation of *ADC* [50]. On average, the brain shows relatively low diffusivity (<5 mm^2^/s) compared to most surrounding tissue (>10 mm^2^/s), resulting in a clear high contrast between the brain and the surrounding tissue. Figure 3C shows an *ADC* map of the brain region from the central imaging slice. In addition to excellent contrast between various brain regions, the CSF system is clearly visible, allowing for the identification of several ventricles. Brain structures, including the diffusive nucleus of the inferior lobe, the ventral zone of the periventricular hypothalamus, and the medial zone of the dorsal telencephalon, show relatively high diffusivity. On the other hand, structures such as the olfactory bulb, cerebellar corpus, longitudinal torus, and medial longitudinal fascicle show relatively low diffusivity. Figure 3D shows estimated *ADC* values for selected ROIs obtained at 17.6 T and 28.2 T (see Figure 1A for ROIs). At both magnetic fields, the highest diffusivity is found in the telencephalon and ventral zone of the periventricular hypothalamus, while the lowest diffusivity is found in the olfactory bulb. Some differences in *ADC* values were seen at 28.2 T as compared to 17.6 T. Although *ADC* values are independent of the applied magnetic field strength, they are affected by SNR. Thus, reported differences at 28.2 T and 17.6 T likely originate from differences in SNR [51].

### 2.4. Diffusion Tensor Imaging (DTI)

Figure 4 shows representative 2D DTI results of adult zebrafish brain obtained at 28.2 T. All images show an identical sagittal slice, rostral (left) to caudal (right), acquired at a resolution of 25 μm × 25 μm, and a slice thickness of 200 μm. Figure 4A–C show *D*_∥_, *D*_⊥_, and *MD* maps of the zebrafish brain, respectively. Various structures could be identified by diffusivity-based contrast. The vascular system and specific brain regions, including the telencephalon, diffusive nucleus of the inferior lobe, and the ventral zone of the periventricular hypothalamus, show a relatively high diffusivity. However, the olfactory bulb and the optic tectum show relatively low diffusivity compared to other brain structures. These results are consistent with observations made by DWI results shown in Figure 3. Furthermore, differences in the D_∥_ and D_⊥_ are observed, indicating anisotropic diffusivity effects. Figure 4D shows the fractional anisotropy map, visualising the extent of anisotropic diffusivity in the zebrafish brain. High *FA* values (light colour) indicate anisotropic diffusion effects, while low *FA* values (dark colour) indicate more isotropic diffusion effects. It is well known that WM structures show the highest FA values in the brain due to the ordered structures of its myelinated axon tracts [14]. Due to high *FA* values, WM brain structures show very high image contrast, allowing the identification of various WM structures, including the medial longitudinal fascicle, ansulate commissure, posterior commissure, and optic tract (indicated in Figure 4D). These structures could not be identified by anatomical imaging or DWI (Figure 2 and Figure 3). Figure 4E shows the DEC *FA* colour map of the central sagittal slice of an adult zebrafish brain, obtained by combining directional information of the principal eigenvalues and *FA* maps. The *FA* colour map further confirms the identification of WM structures by FA, showing the rostral–caudal orientation of the medial longitudinal fascicle and the medial–lateral orientation of the ansulate commissure, posterior commissure, and optic tract.

MRI *FA* colour maps display the primary direction of fastest diffusion within each voxel, offering a useful initial indication of where specific white matter structures are located in the brain. dMRI tractography is essential for visualising the continuous trajectories of white matter bundles non-invasively, and it correlates well with angular structures observed in fibre geometries through confocal microscopy analysis [52]. It is known that signal noise and susceptibility artefacts can lead to anatomically incorrect pathways in dMRI tractography, particularly when reconstructing long pathways, as these distortions can mislead the tracking process, resulting in false or fragmented connections [53]. Additionally, an analysis of crossing fibres in dMRI tractography, in comparison to microscopy on histological slices, revealed that fibres intersecting at medium to low angles (<60°) often lead to inaccurately reconstructed tracts [53]. These challenges require careful use of the technique, particularly in quantitative analysis, where such inaccuracies can significantly affect the results. In this study, we focus primarily on the visualisation of white matter structures, emphasising the qualitative aspects of pathway depiction rather than quantitative metrics. Furthermore, to minimise the accumulation of false positive or false negative pathways, we avoid the construction of long fibre tracts.

Figure 5A shows representative super-resolution stTDI tractography maps of 2D DTI data. Basic tractography algorithms are deterministic, which allows fitting the diffusion tensors to the diffusion data by tracking paths through the principal eigenvector of the tensor [54]. However, deterministic algorithms do not allow for crossing or closely passing fibres, potentially leading to non-existing connections [55,56]. dMRI tractography with CSD processes crossing and closely passing fibres, based on the first-order integration over fibre orientation distribution functions (fODFs) [55]. These fODFs are estimated from response functions, in particular, the estimated signal for single-fibre WM. In Figure 5A, CSD was performed by single-shell single-tissue (ssst) algorithms to estimate the response function. The stTDI maps by ssst-CSD show high-resolution fibre tracts, providing directional information beyond the DTI spatial resolution. Similar stTDI maps have been generated from zebrafish brains in a previous DTI study performed at a lower magnetic field of 16.T [34]. However, the DTI measurement was performed on surgically isolated zebrafish brains treated with an MRI contrast agent. In our work, we performed all the measurements of intact zebrafish at 28.2 T without any contrast agent. One of the shortcomings of performing CSD with ssst CSD algorithms is that it generates a track map without distinguishing individual anatomical structures (such as grey matter, CSF, and WM). Thus, potentially producing WM tracking errors and overestimation [35].

To produce tracks exclusively in the WM area, we applied multi-shell-multi-tissue algorithms to obtain stTDI maps (stTDI msmt-CSD). The msmt-CSD was originally developed and optimised for human brain data [37] and has also been applied to mouse brain DTI data [57]. However, it has not yet been applied to the zebrafish brain. In this study, we implemented and successfully obtained the stTDI map by msmt-CSD in a zebrafish brain (Figure 5B). The response function estimation for WM, GM, and CSF obtained with msmt-CSD derived from our DTI data set is shown in Appendix A. As can be seen from Figure 5B, stTDI by msmt-CSD produced tracks exclusively in WM areas. Furthermore, the stTDI map generated by msmt-CSD demonstrates a significant reduction in noise over those obtained by ssst-CSD (Figure 5), leading to improved identification of WM structures. Figure 5C,D show fODFs estimated by ssst and msmt CSD algorithms, respectively, from the optic tectum (see the highlighted area in Figure 5A and 5B). As shown in Figure 5D, the msmt-CSD fODFs have fewer lobes compared to the ssst-CSD shown in Figure 5C. This difference in lobe number indicates a reduced number of possible tracks obtained by the msmt-CSD method. Our results are consistent with earlier reports in human and mouse brains [37,57]. An increased number of tracks with ssst-CSD indicates tracking errors due to overestimation, resulting in false positive WM tracks. Unlike msmt-CSD algorithms, ssst-CSD does not individually characterise WM, GM, and CSF signals, resulting in WM tracking errors and overestimation [35]. The development of the msmt-CSD algorithm revealed that WM, GM, and CSF have distinct dependencies on *b*-values [35], enabling the deconvolution of their individual signals. By filtering GM and CSF signals strongly present in ssst-CSD, the number of false positive WM tracks is reduced with the msmt-CSD algorithm [36,37].

Next, we acquire data using 3D DTI, as it allows for better visualisation and quantification of white matter tracts, particularly in the areas with complex fibre orientation. The representative 3D DTI results of adult zebrafish brain, acquired at an isotropic resolution of 35 μm, are shown in Appendix A. From the 3D DTI data, *FA* maps and DEC *FA* colour maps were generated. Furthermore, 5 μm stTDI tractography maps were created from 3D DTI data (Appendix A). Both ssst-CSD and msmt-CSD algorithms were used to estimate the response function. Enhanced resolution for crossing and closely passing fibres was clearly obtained with msmt-CSD as compared to ssst-CSD. Similar to 2D DTI data, visualisation of WM structures by stTDI with msmt-CSD is significantly improved compared to ssst-CSD. 

Figure 6 demonstrates the capability to identify WM structures in the zebrafish brain using 3D super-resolution stTDI msmt-CSD. In the sagittal (Figure 6A), coronal (Figure 6B), and axial (Figure 6C) view, WM structures were identified at high contrast and resolution, which were verified by comparing them with detailed atlases of the zebrafish brain [47,48]. A selection of the commissures, fibre tracts, and nerves that could be assigned are shown. This study shows that DTI at the ultra-high field in conjunction with stTDI msmt-CSD provide great resolution for visualisation of white matter tracts in intact zebrafish without the need to isolate the brain.

## 3. Materials and Methods

### 3.1. Zebrafish Husbandry

The husbandry of adult zebrafish described in this study complied with guidelines from the local animal welfare committee of the university (license numbers: AVD1060020171767 and AVD10600202216175), following the international guidelines specified by the EU Animal Protection Directive 2010/63/EU, and was conducted according to standard protocols (www.zfin.org) as described previously [58]. In this study, adult zebrafish (wild type, male, n = 6) were used. Adult zebrafish aged between 4 and 6 months were euthanised through immobilisation by submersion in ice water (0–4 °C) for at least 10 min following cessation of opercular movement and then fixed in 4% buffered paraformaldehyde (Zinc Formal-Fixx, ThermoShandon, UK) for 4 days and subsequently embedded in perfluoropolyether (Fomblin Y, Solvay Solexis S.P.A.) for MRI measurements.

### 3.2. Magnetic Resonance Imaging

MRI was performed in a 28.2 T (1.2 GHz) and a 17.6 T (750 MHz) vertical bore magnet (Bruker Biospin, Ettlingen, Germany). Both systems were equipped with a MICRO 5 gradient system (G_max_ 3 T/m), a 5 mm birdcage RF coil, and a GREAT 60 gradient power supply. Data acquisition and processing were performed using Paravision 360 v3.3 (Bruker Biospin, Ettlingen, Germany). For all measurements, animals were transferred to 5 mm NMR tubes and embedded in Fomblin. To achieve maximal signal intensity, the position of the brain was aligned to the centre of the RF coil. An overview of the experimental setup is provided in Appendix A. Magnetic field homogeneity was achieved by shimming, up to the second order. A variety of scan protocols was used and optimised for the magnetic field strength of 28.2 T. Identical scan protocols were then used for both MR systems (17.6 T and 28.2 T). For anatomical imaging, a two-dimensional (2D) RARE sequence was used. Images were acquired with an echo time (*TE*) of 5.6 ms, a repetition time (*TR*) of 3000 ms, using 4 segmenting refocusing echoes (RARE factor), and a spatial resolution of 23 × 23 μm. RARE images were obtained at a slice thickness of 100 and 200 μm using 64 and 16 averages, respectively. 

For the estimation of spin–spin relaxation times (*T*_2_), a 2D multi-slice multi-echo (MSME) protocol was used based on the Carr–Purcell–Meiboom–Gill (CPMG) sequence [59]. MSME measurements were performed with 15 echo images per excitation at an echo spacing of 5 ms, a *TR* of 2500 ms, 4 averages, at a spatial resolution of 23 × 23 μm, and a slice thickness of 200 μm. For estimation of the spin–lattice relaxation times (*T*_1_), a 2D RARE protocol at variable repetition times (VTR) was used. Signal intensities were measured at a TR of 300, 606, 967, 1408, 1974, 2767, 4100, and 10,000 ms, with a *TE* of 3 ms and 2 averages, at a spatial resolution of 47 × 47 μm and a slice thickness of 200 μm. 

Diffusion weighted imaging (DWI) was performed with a 2D diffusion-weighted spin–echo sequence (DWI SE) with a *TE* of 20.2 ms, a *TR* of 1000 ms, 4 averages, at a spatial resolution of 23 × 23 μm and a slice thickness of 200 μm. MR signals were measured at *b*-values of 75, 500, 1000, 3500, 5000, and 7500 s/mm^2^. Two-dimensional DTI with echo planar imaging (EPI) was performed at an anisotropic resolution of 25 × 25 μm, at a slice thickness of 200 μm, with a *TE* of 12.4 ms, *TR* of 2000 ms, 32 averages, and an EPI factor of 8. Multi-shell experiments were performed with *b*-values of 4, 1000, 3500, and 6000 s/mm^2^ with 8, 12, 24, and 36 directions, respectively. Three-dimensional (3D) DTI was performed at an isotropic spatial resolution of 35 μm, with a *TE* of 9.1 ms, *TR* of 2000 ms, 4 averages, and an EPI factor of 8. Multi-shell experiments were performed with *b*-values of 100, 1000, and 2500 s/mm^2^ with 4, 12, and 24 directions, respectively. Automatic drift compensation was applied to compensate for possible *B*_0_ drift.

### 3.3. Data Processing

Identification of brain regions and WM structures was performed based on the adult zebrafish brain atlas [45], the topological atlas [46], and Ullmann et al. [34]. 

For the estimation of SNR, RARE intensity images were processed in Matlab (mathworks.com). Brain tissue was selected using the volume segmenter tool, and the average signal intensity was calculated. For noise, 10 × 10 voxels were selected outside the fish. From these data, SNR was calculated using Equation (1) [60]:SNR = (μ_S_ − μ_N_)/σ_N_(1)
where μ_S_ is the mean intensity of the brain signal, μ_N_ is the mean intensity of the noise, and σ_N_ is the standard deviation of the noise. SNR maps were created by estimating SNR for each voxel inside the RARE intensity image. Signals outside the fish were set to zero by a signal intensity threshold. 

For the estimation of spin–lattice relaxation times (*T*_1_), spin–spin relaxation times (*T*_2_), and *ADC* values, nine regions of interest (ROIs) were manually selected in the zebrafish brain using Paravision 360 v3.3. *T*_1_ values were estimated by the image sequence analysis tool in Paravision using a nonlinear least square algorithm for the mono-exponential fit function Equation (2) using RAREVTR data:I_t_ = A + I_0_ ∙ (1 − exp (−*t*/*T*_1_))(2)
where I_t_ is the signal intensity at time *t*, A is the absolute bias, I_0_ is the signal intensity at time *t*_0_, *t* is the repetition time (ms), and *T*_1_ is longitudinal relaxation time (ms). From MSME data, *T*_2_ values were estimated using the image sequence analysis tool in Paravision with a nonlinear least square algorithm for the mono-exponential fit function Equation (3):I_t_ = A + I_0_ ∙ exp (−t/T_2_) (3)
where I_t_ is the signal intensity at time *t*, A is the absolute bias, I_0_ is the signal intensity at time *t*_0_, *t* is the echo time (ms), and *T*_2_ is the transverse relaxation time (ms). To reduce the effect of imperfect 180° pulses, uneven echoes were excluded from the data fitting [61]. *ADC* values were estimated from DWI data by the image sequence analysis tool in Paravision with a nonlinear least square algorithm for the mono-exponential fit function Equation (4):I_b_ = A + I_0_ ∙ exp (−b · ADC) (4)
where I_b_ is the signal intensity at *b*, A is the absolute bias, I_0_ is the signal intensity at *b* = 0 mm^2^/s, *b* is the *b*-value (s/mm^2^), and *ADC* is the apparent diffusion coefficient (mm^2^/s). During the estimation of *ADC*, high *b*-values (>2000 s/mm^2^) were excluded from the data fitting. It is well documented that high *b*-values are influenced by non-Gaussian diffusion effects, affecting estimated *ADC* [50]. 

Processing of 2D and 3D DTI data was performed by MRtrix3 [62]. Brain masks were manually created using the volume segmenter tool in Matlab. Using MRtrix3 functions, the denoising of DTI data was based on the random matrix theory [63]. Diffusion tensors were calculated by fitting the diffusion tensor to the log of the denoised DTI data by minimising the weighted least squares and iterated weighted least squares [64,65]. From the tensors, the *D*_∥_, *D*_⊥_, *MD*, and *FA* were calculated. *FA* colour maps were generated by adding the directionality of the principal eigenvalue to *FA* maps. MRtrix3 was used to generate super-resolution (5 μm) stTDI maps from 2D and 3D DTI data. Constrained spherical deconvolution (CSD) was used to resolve crossing fibres by estimating fibre orientation distribution functions (fODFs). Single-shell single-tissue (ssst) CSD response functions were estimated by the Tournier algorithm [66], using the highest b-value of the DTI dataset. fODFs were calculated with the ssst-CSD algorithm [67]. Multi-shell multi-tissue (msmt) CSD response functions were estimated by the Dhollander algorithm [36]. Crude segmentation of WM and GM/CSF voxels in 2D and 3D DTI data was performed at *FA* = 0.2 (default). WM, GM, and CSF voxel selection of 3D DTI data for response function estimation was performed with the top 0.5% of refined WM voxels, 2.0% of refined GM voxels, and 10% of refined GM voxels. For the 2D DTI data, 10% of the refined WM voxels were used in the final selection due to the lower total number of available voxels compared to the 3D DTI data set. Finally, msmt-CSD fODFs were calculated with the msmt-CSD algorithm [37]. Whole brain fibre tracking was performed with the iFOD1 algorithm [68]. One million (2D DTI data) and ten million (3D DTI data) tracks were generated with a minimal track length of twice the voxel size and a maximal track length of ten times the voxel size. Fibre generation was automatically terminated once leaving the pre-defined brain mask. Additional parameters were used in their default mode [56]. Tractography results were transformed into super-resolution track density imaging (TDI) [69], resulting in approximately 5μm isotropic resolution. The directionality of generated tracks is visualised by DEC.

## 4. Conclusions

In conclusion, here we have shown the potential of MRI techniques at ultra-high magnetic fields (28.2 T) to study the zebrafish brain non-invasively. Excellent contrast and SNR were obtained, allowing for the identification of brain structures, as well as fine white matter structures in intact zebrafish brains. Furthermore, ultra-high field DTI was capable of generating reliable and quantitative representations of the fibre organisation within the small brains of zebrafish. This opens up the possibility of studying disease-related changes in white matter structure with super high resolution in a wide range of zebrafish models of human diseases.

## Figures and Tables

**Figure 1 molecules-29-04637-f001:**
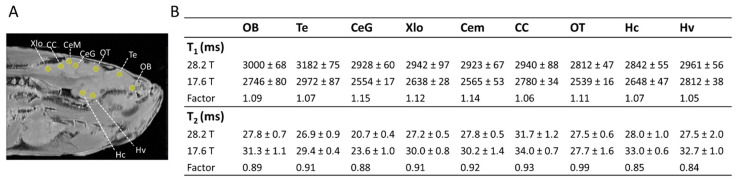
***T*_1_ and *T*_2_ relaxation time measurements in various brain regions of adult zebrafish.** (**A**) Anatomical RARE image of adult zebrafish indicating the location of ROIs selected for *T*_1_ and *T*_2_ quantification. (**B**) *T*_1_ and *T*_2_ relaxation time values in various zebrafish brain regions measured at 28.2 and 17.6 T. The difference between *T*_1_ and *T*_2_ values measured at two different magnetic field strengths is presented as fold change (factor). Selected ROIs: Xlo—Vagal lobe; CC—Cerebellar crest; CeM—Cerebellar corpus, molecular layer; CeG—Cerebellar corpus, granular layer; OT—Optic Tectum; Te—Telencephalon; OB—Olfactory bulb; Hc—Caudal zone of periventricular hypothalamus; Hv—Ventral zone of periventricular hypothalamus. Data represent the mean *T*_1_ and *T*_2_ in ms ± standard error (SE) (error bars); n = 6.

**Figure 2 molecules-29-04637-f002:**
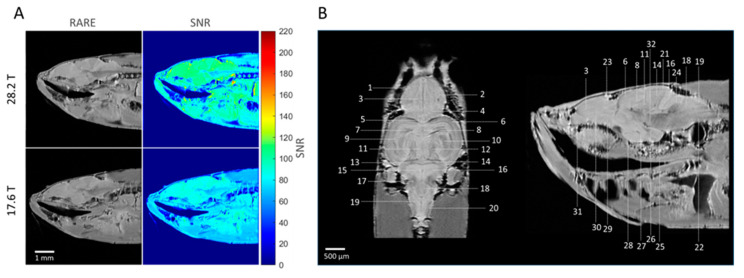
**Comparison of the anatomical images of the zebrafish brain measured at 17.6 T and 28.2 T.** (**A**) Representative RARE images (left column) of the zebrafish brain acquired at 17.6 T and 28.2 T. RARE acquisition details: *TR* 3000 ms, *TE* 5.6 ms, 16 averages, resolution 23 μm × 23 μm × 200 μm, RARE factor 4. SNR maps generated from RARE images (**right column**). (**B**) RARE images of the adult zebrafish brain region in a coronal (**left**) and sagittal (**right**) view acquired at 28.2 T for structure identification. Acquisition details: TR 3000 s, TE 5.6 ms, 64 averages, resolution 23 μm × 23 μm × 100 μm, RARE factor 4. Identified structures are as follows: 1—Central zone of dorsal telencephalon area; 2—Lateral zone of the dorsal telencephalon; 3—Medial zone of dorsal telencephalon; 4—Posterior zone of dorsal telencephalon area; 5—Dorsal habenular nucleus; 6—Optic tectum; 7—Tectal ventricle; 8—Longitudinal torus; 9—Periventricular grey zone of optic tectum; 10—Ventrolateral nucleus of semicircular torus; 11—Medial division of valvula cerebelli, molecular level; 12—Medial division of valvula cerebelli, granular layer; 13—Granular eminence; 14—Cerebellar corpus, granular layer; 15—Rhombencephalic ventricle; 16—Caudal lobe of cerebellum; 17—Medial octavolateralis nucleus; 18—Facial lobe; 19—Vagal lobe; 20—Medial funicular nucleus; 21—Cerebral corpus, molecular layer; 22—Medial longitudinal fascicle; 23—Dorsal sac; 24—Cerebellar crest; 25—Diffuse nucleus of the inferior lobe; 26—Mammillary body; 27—Caudal zone of periventricular hypothalamus; 28—Ventral zone of periventricular hypothalamus; 29—Parvocellular preoptic nucleus, anterior part; 30—Ventral nucleus of ventral telencephalon area; 31—Olfactory bulb; 32—Interpeduncular nucleus.

**Figure 3 molecules-29-04637-f003:**
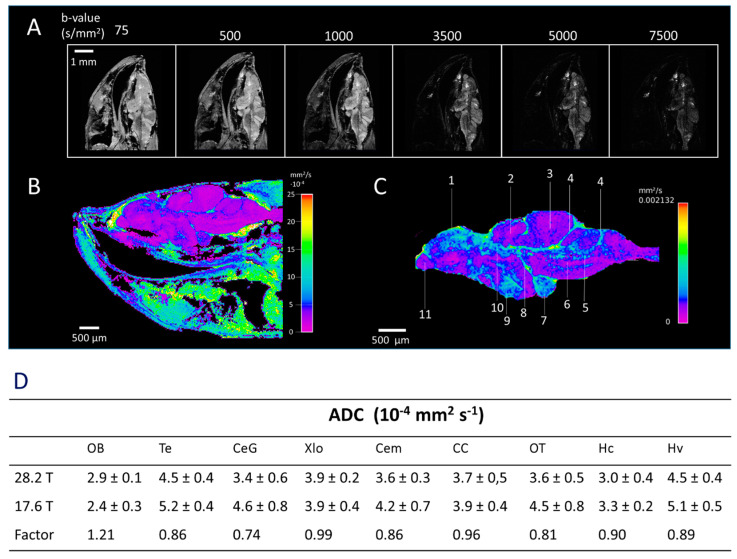
**Diffusion weighted imaging (DWI) of the adult zebrafish brain acquired at 28.2 T.** (**A**) A representative sagittal slice of DWI measurement was taken in the head of a zebrafish at increasing *b*-values. Acquisition details: TR 1000 ms, TE 20.2 ms, 4 averages, resolution 23 μm × 23 μm × 200 μm, and effective *b*-value range 75, 500, 1000, 3500, 5000 and 7500 s/mm^2^. (**B**) Apparent diffusion coefficient (*ADC*) map estimated in the brain area of zebrafish, showing high contrast between brain and surrounding tissue. (**C**) *ADC* map calculated in the brain of zebrafish, demonstrating the identification of several structures based on diffusion characteristics: 1—Medial zone of dorsal telencephalon; 2—Longitudinal torus; 3—Cerebellar corpus; 4—Rhombencephalic ventricle; 5—Ventral rhombencephalic commissure; 6—Medial longitudinal fascicle; 7—Diffusive nucleus of the inferior lobe; 8—Vascular lacuna of area postrema; 9—ventral zone of periventricular hypothalamus; 10—Diencephalic ventricle; 11—Olfactory bulb. (**D**) *ADC* values in various brain regions were acquired at 28.2 T and 17.6 T. For the estimation of *ADC*, solely the effective *b*-value range 75, 500, and 1000 s/mm^2^ was utilised to prevent the influence of non-Gaussian diffusion effects observed at high *b*-values (>2000 s/mm^2^). The difference between *ADC* values measured at two different magnetic field strengths is presented as fold change (factor). Data represent the mean *ADC* (10^−4^ mm^2^ s^−1^) ± standard error (SE) (error bars); n = 6.

**Figure 4 molecules-29-04637-f004:**
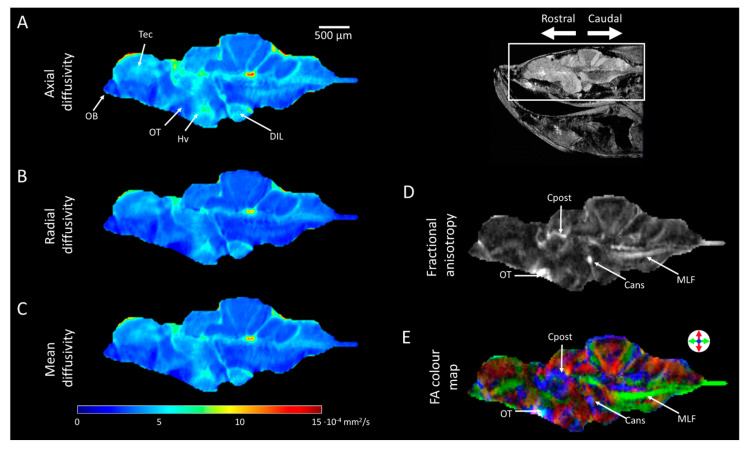
**White matter structure characterisation based on 2D diffusion tensor imaging (DTI) results of adult zebrafish brain acquired at 28.2 T.** A representative central slice in sagittal view showing colour maps of (**A**) axial diffusivity, (**B**) radial diffusivity, and (**C**) mean diffusivity calculated from diffusion tensor data. (**D**) Fractional anisotropy and (**E**) diffusion-encoded-colour (DEC) map of the fractional anisotropy (FA). DEC is used to indicate orientation: green—rostral/caudal, red—dorsal/ventral, and blue—medial/lateral. Abbreviation: MLF—the medial longitudinal fascicle; Cans—ansulate commissure; Cpost—posterior commissure; OT—optic tract.

**Figure 5 molecules-29-04637-f005:**
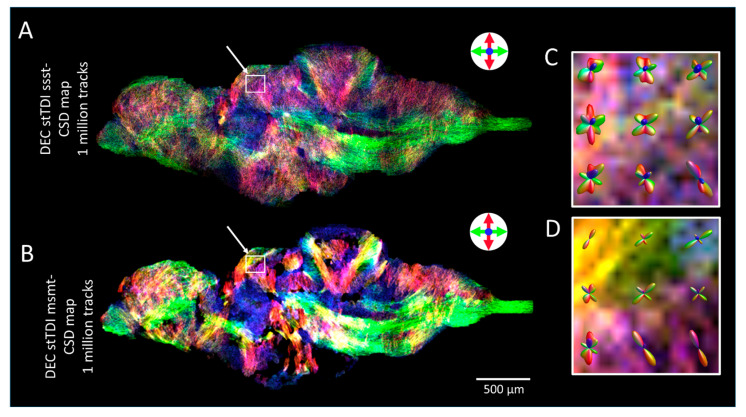
**Whole brain tractography of the zebrafish brain.** (**A**) Super-resolution DEC stTDI map calculated from single-shell single-tissue (ssst) CSD using the tournier algorithm [48] to estimate the response function. (**B**) Super-resolution DEC stTDI map calculated from white-matter response function estimated by multi-shell multi-tissue (msmt) CSD using the dHollander algorithm [35]. Tractography is performed with one million streamlines, a minimal length of twice the voxel size (50 μm) and a maximum length of ten times the voxel size (250 μm). (**C**,**D**) fODFs located in the optic tectum isolated from the 2D stTDI ssst-CSD and msmt-CSD map, respectively. The location of these fODFs is highlighted in (**A**) and (**B**), respectively. DEC is used to indicate orientation: green—rostral/caudal, red—dorsal/ventral, and blue—medial/lateral.

**Figure 6 molecules-29-04637-f006:**
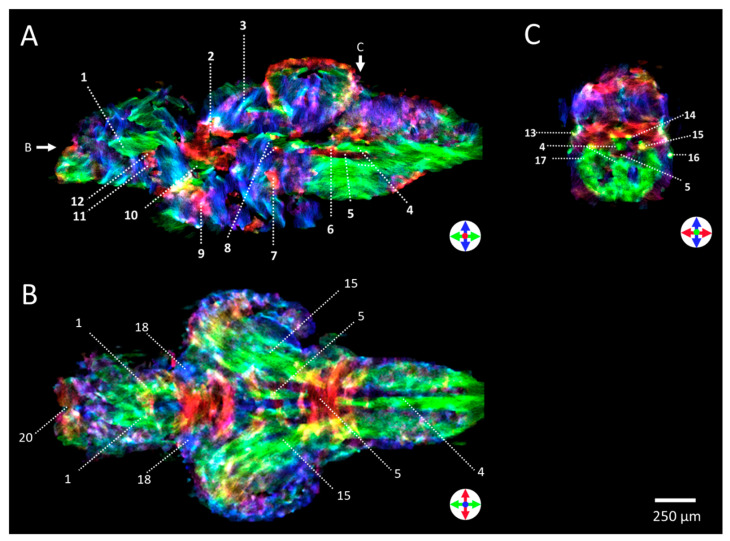
**Three-dimensional super-resolution DEC stTDI msmt-CSD map of the adult zebrafish brain.** The DEC stTDI msmt-CSD maps were generated from 3D DTI data measured at 28.2 T. Shown are representative maps in sagittal (**A**), coronal (**B**), and axial view (**C**). Three-dimensional super-resolution DEC stTDI msmt-CSD map allowed identification of several WM structures in zebrafish brain: 1 (MOT)—Medial olfactory tract; 2 (Cpost)—Posterior commissure; 3 (Ctec)—Tectal commissure; 4 (MLF)—Medial longitudinal fascicle; 5 (Cven)—Ventral rhombencephalic commissure; 6 (IAF)—Inner arcuate fibres; 7 (Cans)—Ansulate commissure; 8 (DiV)—Diencephalic ventricle; 9 (Cpop)—Postoptic commissure; 10 (MFB)—Medial forebrain bundle; 11 (CantV)—Anterior commissure; 12 (CantD)—Anterior commissure; 13 (PLLN)—Posterior lateral line nerve; 14 (VIIs)—Sensory root of the facial nerve; 15 (LLF)—Lateral longitudinal fascicle; 16 (ALLN)—Anterior lateral line nerves; 17 (SGT)—Secondary gustatory tract; 18 (DOT)—dorsomedial optic tract; 19 (Cgus)—Commissure of the secondary gustatory nuclei; 20 (GL)—Glomerular layer of olfactory bulb. DEC is used to indicate orientation: green—rostral/caudal, blue—dorsal/ventral, and red—medial/lateral.

## Data Availability

The data presented in this study are available on request from the author (Y.K).

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
