# Peer review of "Unveiling the Exquisite Microstructural Details in Zebrafish Brain Non-Invasively Using Magnetic Resonance Imaging at 28.2 T"

_molecules, 2024, doi:10.3390/molecules29194637_

Round 1

Reviewer 1 Report

Comments and Suggestions for Authors

The study is methodologically sound and has a clear scope. The manuscript is well-written. I have only minor comments/concerns:

Line 77: The manuscript would benefit from some discussion of the accuracy of tractography for studying brain connectivity (cohort vs single animal). Some discussion of earlier studies using light microscopy to validate tractography based on high field MR microscopy would support this discussion neatly.

Line 133: The stated field dependence of T1 needs backing by citation. Other studies have found linear dependence in human brain, so the power of 1/3 stated here should be commented on. Example reference: https://www.sciencedirect.com/science/article/pii/S1053811920301877?via%3Dihub

Line 215-: What b-values were used for ADC estimation? If too high values are included the non-gaussian effects start influencing ADC estimation. It may also be better use of the data to do diffusion kurtosis analysis if enough directions were acquired.

Line 435: this is not a typical definition of SNR so consider using a standard expression instead (e.g. Brown’s book on MRI).

Author Response

Reviewer 1

The study is methodologically sound and has a clear scope. The manuscript is well-written. I have only minor comments/concerns:

Line 77: The manuscript would benefit from some discussion of the accuracy of tractography for studying brain connectivity (cohort vs single animal). Some discussion of earlier studies using light microscopy to validate tractography based on high field MR microscopy would support this discussion neatly.

***We thank the reviewer for the comments. We have now added discussion regarding the challenges with accuracy of the tractography for studying brain connectivity, especially focusing on challenges with low-angle crossing fibers, MRI noise, and artifacts and their effect on generated fiber tracts. Also, some discussion of earlier studies using light microscopy to validate tractography is added in Result and discussion section.

Line 133: The stated field dependence of T1 needs backing by citation. Other studies have found linear dependence in human brain, so the power of 1/3 stated here should be commented on. Example reference: https://www.sciencedirect.com/science/article/pii/S1053811920301877?via%3Dihub

***A reference to back our statement is added (reference 43), while also recognizing that other references claim the dependency of T1 to B0 is more linearly.

Line 215-: What b-values were used for ADC estimation? If too high values are included the non-gaussian effects start influencing ADC estimation. It may also be better use of the data to do diffusion kurtosis analysis if enough directions were acquired.

*** We are aware that the diffusion-weighted MR signal recorded at high b-values (> 2000 s/mm2 ) are effected by non-Gaussian diffusion effects, influencing estimated ADC. Therefore, the ADC values presented in Fig.3 are estimated solely utilizing the MR signal obtained with the effective b-value range of 75, 500, and 1000 s/mm2. This was not clearly stated in the methodology section and in the discussion. We have now  added it in revised manuscript.

Line 435: this is not a typical definition of SNR so consider using a standard expression instead (e.g. Brown’s book on MRI).

***Thank you for your feedback. We understand that the standard description of SNR in MRI, as outlined in the source you proposed, was not used in our analysis. However, our goal was not to provide a detailed description of the SNR itself, but rather to calculate it specifically for the purpose of comparing the two fields. For this comparison, we utilized the equation as given in our work and previously utilized by Krug et al. (2020) in their SNR tests. We have added this reference to our SNR method section.

Reviewer 2

The paper aims to advance the study of the zebrafish brain by utilizing ultra-high-field MRI at 28.2 T to achieve unprecedented spatial resolution and signal-to-noise ratios. The study demonstrates a 20% improvement in SNR at 28.2 T compared to 17.6 T, which enhances the visualization of brain structures and the establishment of normative T1 and T2 values. Key contributions include the successful application of diffusion tensor imaging (DTI) and tractography at high resolution, as well as the introduction of a novel algorithm for short-track track-density imaging with multi-shell multi-tissue constrained spherical deconvolution, significantly reducing false-positive tracks. These advancements provide detailed, quantitative maps of white matter organization in zebrafish brains, offering a powerful tool for investigating disease-related microstructural changes. Based on the potential to demonstrate microstructural details in Zebrafish brain at 28.2 T. I recommend publication of this manuscript, after the following issues are fully addressed. Minor revision:

***We thank the reviewer for the positive comments.

- The authors provide the experimental materials, imaging sequence parameters, and methodological formulas for data processing in Part III. I suggest that this section can be placed before the experimental results and discussion because the parameters and image results are related. This arrangement will aid readers in better understanding the images and results by providing information upfront.

***We agree with the reviewer, however, according to the format of the Journal “Molecule”, the method section comes after the result section. So, we kept the section order according to the style of the Journal.

- The authors propose using ultrahigh-field small-bore vertical NMR magnets for MRI, which includes a gradient system, a birdcage RF coil, and the zebrafish. This approach is quite novel. Additionally, since the positioning of the zebrafish head is crucial, I recommend including a schematic of the overall experimental setup to provide clearer context.

***We thank reviewer for pointing out the novelty of our experimental setup and for their recommendation to include a schematic overview to the manuscript. We have accordingly added the schematic experimental overview to our supplementary information (Fig. S4).

- In the DWI experiments section, the authors show ADC maps of the zebrafish brain structures and a table of ADC comparisons at 28.2 T and 17.6 T. To better illustrate the advantages of the 28.2 T imaging, I suggest including ADC maps of the zebrafish brain structures at 17.6 T as well. This addition would provide a clearer comparison and highlight the improved contrast achieved at 28.2 T.

***Thanks for your suggestion. The ADC maps of zebrafish brain structures at 17.6 T has been published in our earlier study [ref 38]. In this manuscript we included values in the table to sufficiently highlight the differences between the two field strengths.

Reviewer 2 Report

Comments and Suggestions for Authors

The paper aims to advance the study of the zebrafish brain by utilizing ultra-high-field MRI at 28.2 T to achieve unprecedented spatial resolution and signal-to-noise ratios. The study demonstrates a 20% improvement in SNR at 28.2 T compared to 17.6 T, which enhances the visualization of brain structures and the establishment of normative T1 and T2 values. Key contributions include the successful application of diffusion tensor imaging (DTI) and tractography at high resolution, as well as the introduction of a novel algorithm for short-track track-density imaging with multi-shell multi-tissue constrained spherical deconvolution, significantly reducing false-positive tracks. These advancements provide detailed, quantitative maps of white matter organization in zebrafish brains, offering a powerful tool for investigating disease-related microstructural changes. Based on the potential to demonstrate microstructural details in Zebrafish brain at 28.2 T. I recommend publication of this manuscript, after the following issues are fully addressed.

Minor revision:

- The authors provide the experimental materials, imaging sequence parameters, and methodological formulas for data processing in Part III. I suggest that this section can be placed before the experimental results and discussion because the parameters and image results are related. This arrangement will aid readers in better understanding the images and results by providing information upfront.

- The authors propose using ultrahigh-field small-bore vertical NMR magnets for MRI, which includes a gradient system, a birdcage RF coil, and the zebrafish. This approach is quite novel. Additionally, since the positioning of the zebrafish head is crucial, I recommend including a schematic of the overall experimental setup to provide clearer context.

- In the DWI experiments section, the authors show ADC maps of the zebrafish brain structures and a table of ADC comparisons at 28.2 T and 17.6 T. To better illustrate the advantages of the 28.2 T imaging, I suggest including ADC maps of the zebrafish brain structures at 17.6 T as well. This addition would provide a clearer comparison and highlight the improved contrast achieved at 28.2 T.

Author Response

Reviewer 2

The paper aims to advance the study of the zebrafish brain by utilizing ultra-high-field MRI at 28.2 T to achieve unprecedented spatial resolution and signal-to-noise ratios. The study demonstrates a 20% improvement in SNR at 28.2 T compared to 17.6 T, which enhances the visualization of brain structures and the establishment of normative T1 and T2 values. Key contributions include the successful application of diffusion tensor imaging (DTI) and tractography at high resolution, as well as the introduction of a novel algorithm for short-track track-density imaging with multi-shell multi-tissue constrained spherical deconvolution, significantly reducing false-positive tracks. These advancements provide detailed, quantitative maps of white matter organization in zebrafish brains, offering a powerful tool for investigating disease-related microstructural changes. Based on the potential to demonstrate microstructural details in Zebrafish brain at 28.2 T. I recommend publication of this manuscript, after the following issues are fully addressed. Minor revision:

***We thank the reviewer for the positive comments.

- The authors provide the experimental materials, imaging sequence parameters, and methodological formulas for data processing in Part III. I suggest that this section can be placed before the experimental results and discussion because the parameters and image results are related. This arrangement will aid readers in better understanding the images and results by providing information upfront.

***We agree with the reviewer, however, according to the format of the Journal “Molecule”, the method section comes after the result section. So, we kept the section order according to the style of the Journal.

- The authors propose using ultrahigh-field small-bore vertical NMR magnets for MRI, which includes a gradient system, a birdcage RF coil, and the zebrafish. This approach is quite novel. Additionally, since the positioning of the zebrafish head is crucial, I recommend including a schematic of the overall experimental setup to provide clearer context.

***We thank reviewer for pointing out the novelty of our experimental setup and for their recommendation to include a schematic overview to the manuscript. We have accordingly added the schematic experimental overview to our supplementary information (Fig. S4).

- In the DWI experiments section, the authors show ADC maps of the zebrafish brain structures and a table of ADC comparisons at 28.2 T and 17.6 T. To better illustrate the advantages of the 28.2 T imaging, I suggest including ADC maps of the zebrafish brain structures at 17.6 T as well. This addition would provide a clearer comparison and highlight the improved contrast achieved at 28.2 T.

***Thanks for your suggestion. The ADC maps of zebrafish brain structures at 17.6 T has been published in our earlier study [ref 38]. In this manuscript we included values in the table to sufficiently highlight the differences between the two field strengths.